# Posthoc Interpretability of Neural Responses by Grouping Subject Motor Imagery Skills Using CNN-Based Connectivity

**DOI:** 10.3390/s23052750

**Published:** 2023-03-02

**Authors:** Diego Fabian Collazos-Huertas, Andrés Marino Álvarez-Meza, David Augusto Cárdenas-Peña, Germán Albeiro Castaño-Duque, César Germán Castellanos-Domínguez

**Affiliations:** 1Signal Processing and Recognition Group, Universidad Nacional de Colombia, Manizales 170003, Colombia; 2Automatics Research Group, Universidad Tecnológica de Pereria, Pereira 660003, Colombia; 3Cultura de la Calidad en la Educación Research Group, Universidad Nacional de Colombia, Manizales 170003, Colombia

**Keywords:** EEG Network, motor imagery, motor skills, functional connectivity, class activation maps

## Abstract

Motor Imagery (MI) refers to imagining the mental representation of motor movements without overt motor activity, enhancing physical action execution and neural plasticity with potential applications in medical and professional fields like rehabilitation and education. Currently, the most promising approach for implementing the MI paradigm is the Brain-Computer Interface (BCI), which uses Electroencephalogram (EEG) sensors to detect brain activity. However, MI-BCI control depends on a synergy between user skills and EEG signal analysis. Thus, decoding brain neural responses recorded by scalp electrodes poses still challenging due to substantial limitations, such as non-stationarity and poor spatial resolution. Also, an estimated third of people need more skills to accurately perform MI tasks, leading to underperforming MI-BCI systems. As a strategy to deal with BCI-Inefficiency, this study identifies subjects with poor motor performance at the early stages of BCI training by assessing and interpreting the neural responses elicited by MI across the evaluated subject set. Using connectivity features extracted from class activation maps, we propose a Convolutional Neural Network-based framework for learning relevant information from high-dimensional dynamical data to distinguish between MI tasks while preserving the post-hoc interpretability of neural responses. Two approaches deal with inter/intra-subject variability of MI EEG data: (a) Extracting functional connectivity from spatiotemporal class activation maps through a novel kernel-based cross-spectral distribution estimator, (b) Clustering the subjects according to their achieved classifier accuracy, aiming to find common and discriminative patterns of motor skills. According to the validation results obtained on a bi-class database, an average accuracy enhancement of 10% is achieved compared to the baseline EEGNet approach, reducing the number of “poor skill” subjects from 40% to 20%. Overall, the proposed method can be used to help explain brain neural responses even in subjects with deficient MI skills, who have neural responses with high variability and poor EEG-BCI performance.

## 1. Introduction

As an exercise in dynamic simulation, Motor Imagery (MI) involves practicing the representation of a given motor movement in working memory without any overt motor activity. Simulation of MI mechanisms supervised and supported by cognitive systems activate brain neural systems that improve motor learning ability and neural plasticity, having a high potential for numerous medical and professional applications [1,2,3,4,5]. It has been identified as a promising method for enhancing motor proficiency, with significant educational implications. Furthermore, it has also demonstrated potential for rehabilitating children with neurological disorders. disorders [3,6]. Due to the fact that the Media and Information Literacy methodology proposed by UNESCO covers several cognitive competencies, this aspect becomes essential, as highlighted in [7]. Nevertheless, accurate decoding of brain neural responses elicited by MI tasks poses a challenge since spectral/spatial/temporal features reflect the ease/difficulty subjects (MI ability) face when adhering to the mental imagery paradigm. In this scenario, individual motor skills seriously impact the MI system’s execution.

In practice, MI systems have become the most common application in Brain-Computer Interface (BCI) research to enable communication and control over computer applications and external devices directly from brain activity [8]. BCI systems are typically designed to use Electroencephalogram (EEG) sensors due to their non-invasiveness, high time resolution, and relatively low cost [9]. However, despite these advantages, EEG recordings from electrodes have several limitations, including non-stationarity, a low signal-to-noise ratio (SNR), and poor spatial resolution, among other reported challenges [10]. Thus, effective control of MI-BCI depends on a careful balance between user skills and EEG-montage constraints [11]. Unfortunately, only a small number of individuals possess the necessary skills to control their BCIs effectively. Moreover, an estimated 15–30% of people lack the skills needed to complete MI tasks accurately, resulting in poor performance (known as BCI-Inefficiency) [12]. As a result, the application of the imagery paradigm is often limited to laboratory settings, as the EEG-based signal analysis and implementation of BCIs with acceptable reliability have become increasingly complex procedures.

A growing body of research has focused on developing strategies to address BCI-Inefficiency, with several promising approaches emerging. One such approach is to adapt the MI BCI system to specific frameworks of application, as demonstrated by recent studies [13,14,15,16]. In addition, these studies have explored various ways to optimize the performance of MI, including by customizing the system’s parameters to suit the user’s needs and abilities. Another promising strategy involves developing personalized systems that account for individual cognitive variability [17,18]. Considering factors such as attentional capacity, memory, and learning styles, these personalized systems can help improve the effectiveness of MI-BCI control, even for those with limited motor skills. Likewise, identifying subjects with a poor motor performance at the early stages of BCI training is another essential strategy for improving MI efficiency [19]. By identifying those who may struggle with BCI control early on, interventions can be implemented to help these individuals develop the necessary skills and optimize their performance.

In turn, determining the lack of motor skills at an early stage of EEG BCI system training implies assessing and interpreting the neural responses elicited by MI across the considered subject set. Yet, MI responses may depend on factors like brain structure variability, diversity of cognitive strategy, and earned individual expertise [12]. A widely-used approach to deal with EEG variability is developing BCI processing pipelines with optimized spatial pattern transformations that may account for conventional approaches [20,21]. Deep learning (DL) methods, especially Convolutional Neural Networks (CNN), have been investigated to optimize signal analysis since they automatically extract dynamic data streams from neural responses to address the complexity of EEG data associated with MI tasks and the variability of users [22,23,24]. Nevertheless, DL models generate high-level abstract features processed by black boxes with meaningless neuro-physiological explainability, resulting in a severe disadvantage in ensuring adequate reliability compared to their high performance [25].

Compared to other deep learning models, CNN-based classifiers offer several key advantages, including the ability to learn end-to-end from raw data and requiring fewer tuning parameters. Besides, end-to-end learning allows for a more streamlined and automated process, reducing the need for manual feature engineering and enabling the model to learn features directly from the data [26,27,28,29]. Moreover, visualization of feature maps from convolutional layers or class-discriminative features computed by the Class Activation Maps (CAM) is among the most used technique for a post-hoc explanation of well-trained CNN learning models [30,31,32]. Generally, different approaches to visualizing the learned CNN filters fall into two classes: extracting trained weights at a selected layer and from reduced low-dimensional representations weighted across the layers, like developed for Motor Imagery in [33,34]. To better encode the diversity of stochastic behaviors, the reduced feature representations are clustered, separating the individuals into groups according to selected metrics of MI BCI performance, like developed in [35]. Despite this, the widespread use of DL models is prevented in functional neuroimaging because of the difficulty of training on high-dimensional low-sample-size datasets and the unclear relevance of the resulting predictive markers [36]. Therefore, CNN-based EEG decoding should be improved by verifying the learned weights’ discriminating capability and relationship to neurophysiological features, as stated in [37].

This work presents a CNN framework to learn MI patterns from high-dimensional EEG dynamical data. The latter is achieved while ensuring that the interpretability of the neural responses is maintained through the use of connectivity features extracted from class activation maps. Also, to address the issue of inter/intra-subject variability in MI-EEG data, our work adopts two different approaches. The first approach involves the extraction of functional connectivity from spatiotemporal CAMs. Notably, we coupled the well-known EEGnet architecture and a novel kernel-based cross-spectral distribution estimator founded on a Gaussian Functional Connectivity (GFC) measure. This approach helps to overcome the limitations of traditional connectivity measures by capturing both the linear and non-linear relationships between the different brain regions. The second strategy involves clustering the subjects based on their achieved classifier accuracy and relevant channel-based connectivities. This clustering aims to identify common and discriminative patterns of motor skills. By doing so, the proposed approach can better account for the inter-subject variability often encountered in MI-EEG data. The validation results from a bi-class database demonstrate that the proposed approach effectively enhances the physiological explanation of brain neural responses. This is particularly true for subjects with poor MI skills and those with high variability and deficient EEG BCI performance. Overall, the proposed method represents a significant contribution to the field of neural response interpretation, and it has the potential to facilitate the development of more effective and accurate BCI systems. Of note, the present work continues our framework in [38] by eliminating the characterization stage to make the developed deep model take advantage of big data. A more complex database with more subjects is then used to evaluate the resulting end-to-end model. Furthermore, we introduce a CNN framework to learn from high-dimensional dynamical data while maintaining post-hoc interpretability using connectivity features extracted from class activation maps.

The agenda is as follows: Section 2 defines the used EEG-Net-based Classification framework of MI tasks, the Score-Weighted Visual Class Activation Maps estimated from EEG-Net, the GFC fundamentals, and the two-sample Kolmogorov-Smirnov test used to ensure the significance of extracted functional connectivity measures. Section 3 describes the validated EEG data set and the parameter setting of the CNN training framework. Section 4 explains the classification results of CAM-based EEGnet Mask, clustering of Motor Imagery Neural Responses using Individual GFC measures, and the Enhanced Interpretability from GFC patterns according to the clusterized motor skills. Lastly, Section 5 gives critical insights into their supplied performance and addresses some limitations and possibilities of the presented approach.

## 2. Materials and Methods

We present the fundamentals of EEGnet-based discrimination and the approach of CAM using a GFC-based EEG representation to improve the posthoc DL interpretability of elicited neural responses. In addition, we present a two-sample Kolmogorov-Smirnov test used to prune the GFC-based channel relationships matrix that is performed to enhance individuals’ grouping according to their MI skills.

### 2.1. EEGnet-Based Classification of Motor Imagery Tasks

This convolutional neural network architecture includes a stack of spatial and temporal layers, $\left\{W_{l}\in R^{P_{l-1}\times P_{l}},b_{l}\in R^{P_{l}}\;:\;l\in L\right\}$, fed by the input-output set, $\left\{X_{n}\in R^{C\times T},\lambda_{n}\in {\left\{0,1\right\}}^{K}\;:\;n\in N\right\}$. Here, $P_{l}\in N$ denotes the *l*-th layer unit set, L∈N is the network depth, C∈N is the number of EEG channels, T∈N is the recording length, N∈N is the number of trials captured for each of K∈N MI classes of MI tasks.

The EEGnet model, noted as $M\left(X\right):R^{C\times T}\;\to \;{\left[0,1\right]}^{K}$, performs prediction $\hat{\lambda }\in {\left[0,1\right]}^{K}$ of each one-hot class membership, λ, as below: (1)$$M\left(X\right)=\left({\tilde{X}}_{L}\circ \cdots \circ {\tilde{X}}_{1}\right)\left(X\right)\to \hat{\lambda }$$
where ${\tilde{X}}_{l}=\phi_{l}\left({\tilde{X}}_{l-1}⊗W_{l}+b_{l}\right)\in R^{P_{l}}$ is a feature map holding $P_{l}\in N$ elements at *l*-th layer, $\xi_{l}\;:\;R^{P_{l-1}}\;\to \;R^{P_{l}}$ is a representation learning function, $b_{l}\in R^{P_{l}}$ is a bias term, and $\phi_{l}\left(\cdot \right)$ is a non-linear activation that is fixed as a sigmoid and softmax function for the bi-class and multiclass scenarios, respectively. Notations ∘ and ⊗ stand for function composition and proper tensor operation (i.e., 1D/2D convolution or fully connected-based product), respectively. Of note, we set $\tilde{X_{0}}=X$, ${\tilde{X}}_{L}=\hat{\lambda }$ in the iterative computation of feature maps.

The solution in (Equation 1) implies the parameter set computation of $\Theta =\{W_{l},b_{l}\;:\;l=L\}$, performed within the following minimizing framework: (2)$$\Theta^{*}=\arg\min_{\Theta }E\left\{L\left(\lambda_{n},{\hat{\lambda }}_{n}|\Theta \right)+\gamma \Omega \left(\Theta \right):\forall n\in N\right\},$$
where $L:{\left\{0,1\right\}}^{K}\times {\left[0,1\right]}^{K}\to R$ is a given loss function, Ω(·) is a regularization function, $\gamma \in R^{+}$ is a trade-off, while E{·} stands for the expectation operator.

In order to support EEG classification, the compact EEGNet convolutional network minimizes (Equation 2), providing convolutional kernel connectivity between inputs and output feature maps that can apply to different MI paradigms. As seen in Figure 1, the EEGNet pipeline begins with a temporal convolution to learn frequency filters, followed by a depth-wise convolution connected to each feature map to learn frequency-specific spatial filters. Further, the separable convolution combines a depthwise convolution, which learns a temporal summary for each feature map individually, followed by a point-wise convolution, which learns how to mix the feature maps optimally for class-membership prediction.

### 2.2. Score-Weighted Visual Class Activation Maps from EEG-Net

To highlight the most relevant input features, we conduct a CAM-based relevance analysis over the EEG-Net pipeline to favor the post-hoc interpretability of those relevant spatio-temporal brain patterns that contribute the most to discriminating the elicited MI neural responses. To this end, we employ the Score-Weighted Visual Explanation approach, termed Score-CAM, that eliminates the dependence on gradient-based interpretability methods through the EEGNet network forward passing score on target class and a further linear combination of relevance weights. Thus, we extract the upsampled class-specific CAM matrix, $A_{l}\left({\hat{\lambda }}_{k}\right)\in R^{C\times T}$, that regards *k*-th estimated label, ${\hat{\lambda }}_{k}\in \left[0,1\right]$, as follows [39]: (3)$$A_{l}\left({\hat{\lambda }}_{k}\right)=\bar{\mu }\left(\sum_{\forall d\in D_{l}}\alpha_{l}^{d}\left({\hat{\lambda }}_{k}\right){\tilde{X}}_{l}^{d}\right),\;\forall k\in K$$
where $\bar{\mu }\left(\cdot \right)$ is the resulting function composition between the upsampling function μ(·) and the activation function in the form ReLU{η}=max(0,η)∀η∈R, the matrix ${\tilde{X}}_{l}^{d}\in R^{P_{l}}$ computed on the trial basis, sizing $P_{l}\;=\;C_{l}\times T_{l}$, holds *d*-th feature map at *l* layer, $D_{l}\in N$ is the number of filters implementing the 1D/2D convolutional operations by the M(·) architecture, and $\alpha_{l}^{d}\left({\hat{\lambda }}_{k}\right)\in \left[0,1\right]$ is a combination weight that is computed as follows: (4)$$\alpha_{l}^{d}\left({\hat{\lambda }}_{k}\right)=\frac{{\tilde{\alpha }}_{l}^{d}\left({\hat{\lambda }}_{k}\right)}{\sum_{d=1}^{D_{l}}{\tilde{\alpha }}_{l}^{d}\left({\hat{\lambda }}_{k}\right)},$$
where the estimate ${\tilde{\alpha }}_{l}^{d}\left({\hat{\lambda }}_{k}\right)\in R^{+}$ is computed in a post-hoc way (namely, after the EEG-Net training) by *k*-th class prediction M(·|k) as ${\tilde{\alpha }}_{l}^{d}\left({\hat{\lambda }}_{k}\right)\;=\;M\left(X⊙\mu \left({\tilde{X}}_{l}^{d}\right)|k\right)-M\left(X|k\right)$, where notation ⊙ stands for the Hadamard product. Note that the Score-CAM representation is constrained by ReLU-based thresholding in (Equation 3) to assemble positive definite values of relevance into $A_{l}\left({\hat{\lambda }}_{k}\right)$, implying discriminant EEG inputs that increase the output neuron’s activation rather than suppressing behaviors [40].

Consequently, the weights estimated in Equation (Equation 4) support the CAM-based approach to boost Spatio-temporal EEG patterns, represented by the element-wise feature maps (or masks), which are salient in terms of discriminating between MI labels.

### 2.3. Pruned Gaussian Functional Connectivity from Score-CAM

Consider two Score-CAM-based EEG records $a,a^{\prime }\in A$ for a given EEGNet layer *l* and predicted class ${\hat{\lambda }}_{k}$. Their correlation can be expressed as a generalized, stationary kernel, $\kappa :R^{T}\times \;R^{T}\to R$, if their spectral representations meet the following assumption [41]:(5)$$\kappa \left(a-a^{\prime }\right)=\int_{\tilde{f}\in \Xi }\exp\left(j2\pi {\left(a-a^{\prime }\right)}^{⊤}\tilde{f}\right){\tilde{S}}_{aa^{\prime }}\left(\tilde{f}\right)d\tilde{f},$$
where $\tilde{f}\in \Xi$ is a vector in the spectral domain and ${\tilde{S}}_{aa^{\prime }}\left(\tilde{f}\right)\in C$ is the cross-spectral density function with $\tilde{S}aa^{\prime }\left(\tilde{f}\right)=dP_{aa^{\prime }}\left(\tilde{f}\right)/d\tilde{f}$, and $P_{aa^{\prime }}\left(\tilde{f}\right)\in \left[0,1\right]$ is the cross-spectral distribution. The cross-spectral distribution between CAM-based records in Equation (Equation 5) can be computed as follows [42]:(6)$$P_{aa^{\prime }}\left(\Xi \right)=2\int_{\tilde{f}\in \Xi }F\left\{\kappa \left(a-a^{\prime }\right)\right\}d\tilde{f}.$$
F{·} stands for the Fourier transform. Then, nonlinear interactions between CAM-based boosted EEG channels are coded for a more accurate depiction of neural activity.

Of note, a stationary kernel is employed to preserve temporal dynamics of EEG signals, essential for MI classification accuracy. Consequently, the Gaussian kernel is favored in pattern analysis and machine learning for its versatility in approximating functions and its mathematical tractability [43]. These properties make it a prime candidate for calculating the Kernel-based Cross-Spectral Functional Connectivity in Equation (Equation 6), fixing the pairwise relationships between a and $a^{\prime }$ as: (7)$$\kappa_{G}\left(a-a^{\prime };\sigma \right)=\exp\left(\frac{-{∥a-a^{\prime }∥}_{2}^{2}}{2\sigma^{2}}\right),$$
where ${∥\cdot ∥}_{2}$ is the L2 distance and $\sigma \in R^{+}$ is a given bandwidth hyperparameter, commonly fixed based on the median of the input distances.

Finally, to prune the Gaussian Functional Connectivity (GFC), as in Equations (Equation 6) and (Equation 7), we need to determine which connections are crucial for class separation. A high correlation in the functional connectivity matrix does not necessarily lead to higher class separation, so we use the two-sample Kolmogorov–Smirnov (2KS) test, as in [44], to address this. The null hypothesis is that both samples come from the same distribution. In turn, we group trials for each CAM GFC connection for a subject by label to form samples representing class membership. Each pair is then tested with the 2KS test and connections with *p*-values ≤ 0.05 are kept, implying the samples are from different distributions and the classes are separable. Finally, a *p*-value matrix indicating relevant connections is generated.

### 2.4. Post-Hoc Grouping of Subject Motor Imagery Skills

We evaluate the proposed approach for improving the post-hoc interpretability of the Deep Learning-based model using the Class Activation Maps and the pruned Gaussian Functional Connectivity to characterize the inter/intra-subject variability of the brain signals. This purpose is addressed within the evaluation pipeline shown in Figure 2, including the following stages: (i) Preprocessing of raw EEG data to be fed the DL model; (ii) Bi-class discrimination of motor-evoked tasks within a shallow ConvNets framework (EEGNet); (iii) In the post-hoc analysis, we compute the CAMs over the Conv2D layer, followed by a clustering analysis performed on a reduced set of extracted features, using the well-known t-distributed Stochastic Neighbor Embedding (t-SNE), to enhance the explainability of produced neural responses regarding motor skills among subjects Also, pruned GFC is used for visual inspection results and clustering enhancement.

### 2.5. GigaScience Database

In this collection, available at http://gigadb.org/dataset/100295, accessed on 1 October 2022, EEG data from fifty (50) subjects have been assembled according to the experimental paradigm for MI, as shown in Figure 3. The MI task execution starts with a fixation cross appearing on a black screen within 2 s. Afterward, for 3 s, a cue instruction appeared on the screen, requesting that each subject imagine the finger movements (initially, with the forefinger and then the small finger touching their thumb). Then, a blank screen appeared at the start of the break, lasting randomly for a short period between 4.1 to 4.8 s. These procedures were repeated 20 times for each MI class in a single testing run, collecting 100 trials per individual (each one lasting T=7 s) for either labeled task (left hand or right hand). Data were acquired at 512 Hz sampling rate by a 10-10 placement using an *C*-electrode montage (C=64), see Figure 4.

## 3. Experimental Set-Up

### 3.1. Parameter Setting of Trained CNN Framework

In the preprocessing stage, each raw channel is passed through a bandpass filter within [4–40] Hz using a five-order Butterworth filter, as suggested in [45]. To enhance the physiological interpretation of implemented experimental paradigm, we analyze the dynamic at the representative time segment from 2.5 s to 5 s (i.e., motor imagery interval). Further, the EEG signal set is standardized across the channels with the *z*-score approach.

Afterward, the resulting preprocessed EEG multi-channel data feeds the EEGNet framework with a shallow ConvNet architecture. The CNN model is implemented by downsampling all cropped EEG recordings to 128 for feeding the EEGNet model, fixing the parameter set to the values as displayed in Table 1. The CNN learner includes a categorical cross-entropy loss function and the Adam optimizer.

Table 1 brings a detailed description of the employed network architecture for which the following values are fixed: The number of band-pass feature sets extracted from EEG data by a set of 2D convolutional filters $F_{1}\;=\;16$; 2D CNN filter length is 128 Hz (i.e., half of the sampling rate); and the number of temporal filters along with their associated spatial filters is adjusted to $F_{2}\;=\;DF_{1}$ in SeparableConv2D. Besides, validation is performed by running 100 training iterations (epochs for performing validation stopping), storing the model weights that reach the lowest loss. The model validation is conducted on a GPU device at a Google Collaboratory session, using TensorFlow and the Keras API.

The final step was to perform a group analysis of individuals using the *K*-means approach for clustering the performance metrics and Kolmogorov test values extracted from connectivity measures. Note that the feature set is *z*-scored to feed the partition algorithm. Besides, we apply the well-known t-distributed Stochastic Neighbor Embedding (t-SNE) for dimensionality reduction [46,47], selecting the first two components.

### 3.2. Quality Assessment

The proposed methodology is evaluated in terms of the EGGNet binary classification estimated through the cross-validation strategy, which consists of prearranging two sets of points: 80% for training and the remaining 20% for hold-out. The following metrics assess the classifier performance [47]: accuracy, kappa value, f1-score, precision, and recall. Besides, a Nemenyi post-hoc statistical test is performed at a significance level of 0.05 [48]. The latter aims to compare subject-dependent EEGNet baseline performance vs. our CAM-based enhancement.

## 4. Results and Discussion

### 4.1. Classification Results of CAM-Based EEGnet Masks

Figure 5 displays the classification accuracy results obtained by the EEGnet model (colored with a blue line) with parameters tuned as described above. In order to better understand the values, we rank them in decreasing order of mean accuracy to reveal that more than half of them fall below the 70% level (horizontal red line) and may need more MI skills. Otherwise, a system with so many poor-performing individuals makes MI training ineffective.

Results for individual masks are also displayed (colored with an orange line) and suggest the improved performance of the classifier. The EEGNet approach enhanced by CAM-based representations is thus more effective at dealing with hardly discernible neural responses, reducing the number of subjects labeled as inadequate in MI from 21 (∼40%) to 11 (∼20%). Furthermore, concerning the Nemenyi post-hoc statistical test, the results demonstrate a significant difference between our CAM-based enhancement and the benchmark EEGNet discrimination approach. Specifically, this difference was most pronounced in subjects with lower performance levels. These findings highlight the potential of our proposed approach to improve the accuracy and reliability of EEG-based analyses, particularly for individuals with poorer MI skills.

A mapping of GFC connectivity based on CAM-based masks is also shown to understand individual neural responses better. For illustration purposes, the topoplot representations extracted from GFC connectivity maps are contrasted for the most successful and poorest performing subjects (#43 and #51, respectively). The displayed topoplot of each subject in Figure 6 shows high response amplitudes distributed evenly over the Frontal and central areas, as expected for the neural responses elicited by MI paradigms [49]. There are, however, substantial differences between both subjects in the assessed relevant GFC links.

A notable finding is that the neural responses in the best-performing individual produced much lower discriminant links, primarily concentrated in the Sensory-Motor region. At the same time, the discriminative links of #43 are more related to the contralateral hemisphere. There may be an explanation for this effect if the skilled subject is more focused on performing tasks based on the MI paradigm.

By contrast, neural connections throughout the brain in the subject with poor performance are twice as high as in the skilled individual. Furthermore, the relevant links of #51 include associations between occipital and frontal areas in the cerebral cortex with channels not primarily related to the imagination of motor activities, suggesting a spurious electrophysiological mechanism during MI performance. Because of these factors, poor-performing subjects tend to be less accurate in distinguishing between MI tasks [50].

Based on the above information, we calculate the connectivity matrix after applying a two-sample Kolmogorov-Smirnov test, as carried out in [51] for the goodness of fit. As can be observed in Figure 7, the best-performing subject produces most of the statistically significant connections all above the sensory motor region following the MI paradigm. In contrast, the poorest-performing subject holds low statistically significant links, making the activity over the motor cortex barely distinguishable from the remaining brain areas.

### 4.2. Clustering of Motor Imagery Neural Responses Using Individual GFC Measures

As inputs to the partition algorithm, we include several classifier performance metrics since we assume that the more distinguishable the elicited neural responses between each labeled task, the better the skills of a subject in performing the MI paradigm. In addition to the EEG-Net classifier metrics, the Kolmogorov test values are also evaluated and assessed for the Gaussian FC measures of the individual ScoreCAMs. For encoding the stochastic behavior of metrics, the clustering algorithm is fed by their reduced space representation using the t-SNE projection, as described in [42].

As shown in the left scatter plots of Figure 8, we performed the *k*-means algorithm on the reduced feature set for each MI task, separating the individuals into three groups according to their motor skills: The best-performing subjects (termed Group I and colored in green); The partition with intermediate MI abilities (Group II in yellow); Group III (red color) with the worst-performing individuals. We separately analyze both Motor Imagery Tasks due to Hand Dominance influence, as discussed in [52]. The grouping quality is assessed by the Silhouette coefficient displayed in the center plots, showing differences in the partition density: the more inadequate the motor skills, the more spread the partition. A further aspect of evaluation is the effect of CAM masks on enhanced classifier performance, which increases as the individual’s MI skills decrease (see right-sidebar plots).

By coloring the squares into three clusters, Figure 9 enhances the explanation of the extracted connectivity measures with statistical significance and shows that the partition density of best-performing subjects (green square) by both MI tasks is alike and well clustered, reaching high silhouette values. For GII, three subjects are outliers, making this fact more evident for the data point labeled as #20, especially in the right-hand task. This result may be explained since this subject holds very few trials.

Lastly, there is a tendency for the GIII cluster to fragment into smaller partitions. This issue is clarified by recalculating the clustering algorithm and exploring the number of subgroups within k={3,4,5,6}. According to the right-side column, fixing k=3 provides the highest silhouette value, being more evident in the scatter plot estimated for the right-hand task. Apparently, the MI paradigm is performed quite distinctly by each GIII individual, thus worsening the classifier accuracy.

### 4.3. Enhanced Interpretability from GFC Patterns According to Clusterized Motor Skills

To bring a physiological foundation to clustered motor skills, we analyzed the brain neural responses elicited by MI paradigms through the GFC patterns extracted from the Score-based CAMs and averaged across each subject partition. As explained above, we compute the pairwise functional connectivity graph that contributes the most to discriminating between the labeled MI-related tasks, selecting electrode links that fall within the 90-percentile of normalized relevance weight calculations, as shown in Figure 10. However, two analysis scenarios are considered: The statistically significant connections computed by the Komolgorov-Smirnov test (see Figure 7) and the connectivity set without passing under the hypothesis test.

The top and middle rows in the former scenario display the connectograms computed for each MI task, respectively. As can be seen, each Group of individual skills triggers different sets of connections in terms of nodes and relevant links, which may lead to specific network integration and segregation changes, affecting the system’s accuracy. Accordingly, the connectivity graphs estimated for the Group of the best-skilled subjects hold high evoked response amplitudes densely spread over the Frontal, Central, and Sensory-Motor areas involved in motor imagery. At the same time, GII yields a connectivity set similar to GI, with most links spreading over the primary motor cortex. However, several EEG electrodes generate discriminating links over the temporal and parietal-occipital areas, which are assumed uncorrelated with MI responses. This situation worsens in the Group of poor-performing individuals with a link set covering the whole scalp surface, regardless of the MI task. Moreover, the most robust links are elicited over the parietal-occipital cortical areas, which are devoted mainly to visuospatial processes. This finding related to relevant connectivities outside the primary motor cortex may account for the strength of DL models for improving accuracy at the expense of searching for any discriminant patterns of neural activity. Patterns may be found even among spurious artifacts in the data acquisition procedure or related to irregularly elicited activations performed in the MI paradigm.

As shown in the bottom row, the connectivity maps for the second scenario are derived with Kolmogorov-Smirnov boosting masks, resulting in sets of discriminant links generated by electrodes just over the primary motor cortex, regardless of the partition of motor skills. It is worth noting that the connectograms achieved by GI are similar for either considered analysis scenario. The fact that the best-performing subjects can achieve very high accuracy even with simple classifiers is widely known. Nonetheless, the more the accuracy of subject partitions, the more robust links elicited by the MI responses.

## 5. Concluding Remarks

We introduce an approach to improving the post-hoc interpretability of neural responses using connectivity features extracted from class activation maps. The Spatiotemporal CAMs are computed using a CNN model and employed to deal with the inter/intra-subject variability for distinguishing between bi-class motor imagery tasks. Specifically, we extract the functional connectivity between EEG channels through a novel kernel-based cross-spectral distribution estimator to cluster subjects according to their achieved classifier accuracy, aiming to find common and discriminative patterns of motor skills. Based on validation results on a tested bi-class database, the proposed approach proves its usefulness for enhancing the physiological explanation of MI responses, even in cases of inferior performance. Although the approach may facilitate the increased use of neural response classifiers based on deep learning, the following aspects are to be highlighted for its implementation:

Deep-Learning based Classifier Performance. The deep model is implemented through EEGNet, a compact CNN specially developed for EEG-based brain-computer interfaces, which allows for reaching a bi-class accuracy of 69.7%. The additional use of proposed connectivity masks extracted from the electrode contribution increases the accuracy to 78.2% (EEGNet+ScoreCam) on average over the whole subject set. Table 2 delivers the accuracy of a few state-of-the-art approaches with similar discriminative deep learning models reported recently, indicating that the proposed spatiotemporal CAM approach is competitive. The works in [53,54] are higher accurate, though no interpretable machine-learning models are offered. By contrast, the CNN-based approach with a close value to our accuracy is suggested in [55], which performs data augmentation and employs Cropping methods for improving machine learning models, allowing plotting topographic heads for spatial interpretation.

Therefore, a broader class of training procedures should be considered to improve the explainability of CNN-based models. To this extent, several strategies have been suggested recently for raising the classifier performance of CNN in discriminating MI tasks, like enhancing their convergence in [60,61], or extracting data at different time-resolutions [62]; this latter strategy may be a better choice for interpretation.

Enhanced interpretability using CNN-based Connectivity Features. This proposed approach uses functional connectivity with kernel-based cross-spectral distribution estimators to improve the computation of joint and discriminative patterns of EEG channel relationships. These correlations are essential in interpreting patterns of the primary motor area that become active and are elicited by motor imagery tasks. In this regard, Table 2 illustrates the existing balance between desirable high classifier performance and meaningful explainability of results to be achieved by deep learning models.

CAMs have previously been reported for CNN classifiers to discover shared EEG features across subject sets. However, we suggest extracting Functional Connectivity measures instead of using the trained weights (like in [33,63,64]) to identify brain regions contributing most to the classification of MI tasks. A convenient trade-off between interpretability and accuracy is thus achieved.

Clustering of subjects according to their motor skills. This model illustrates a potential approach to enhancing CNN classifiers to discover shared features related to EEG MI tasks across subject sets. The use of CNN models is also increasing when screening subjects with poor skills. In [24], for example, authors aim to detect users who are inefficient in using BCI systems. However, in addition to identifying poor performers, explaining why they cannot produce the desired sensorimotor patterns is essential. On the other hand, the strength of DL models lies in their ability to improve accuracy at the expense of finding any discriminant patterns of neural activity. As a result, DL models may estimate patterns unrelated to the primary motor cortex, i.e., they are responses with no physiological interpretation according to the MI paradigm.

Instead, the proposed method allows for determining the actual spatial locations of neural responses. Thus, we propose clustering motor skills into three partitions: Group I with high accuracy (good performing of MI paradigms and not subject to artifacts), GII with fair accuracy (adequate MI performing but subject to artifacts), and GIII with low accuracy (very variable MI performance of tasks and also subject to artifacts).

As future work, the authors plan to address the issue of high intra-class variability and overfitting, specifically in the worst-performing subjects, by incorporating more powerful functional connectivity estimators, for instance, using regularization techniques based on Rényi’s entropy [65]. Besides, we plan to test more powerful DL frameworks, including attention mechanisms for decoding stochastic dynamics, as suggested in [66]. Further during-training analysis of the inter-class variability should also be conducted, specifically focusing on how each subject performs within different experiment runs. Our goal is to determine if subjects learn MI tasks as they progress through the runs and whether this leads to better performance in subsequent runs. Validation of other MI EEG data will be carried out.

## Figures and Tables

**Figure 1 sensors-23-02750-f001:**
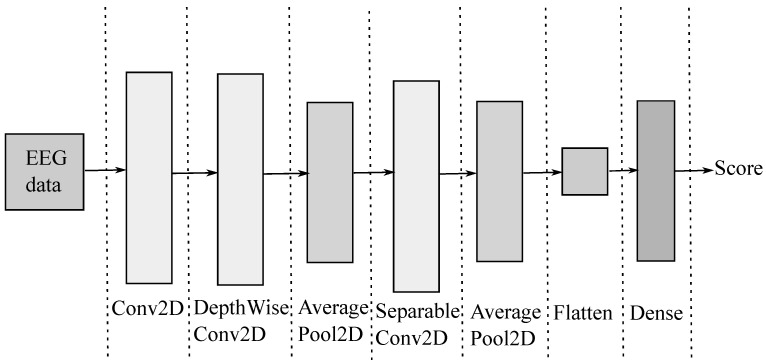
EEGnet main sketch. First column: input EEG. Second column: temporal convolution (filter bank). Third and fourth columns: spatial filtering convolution. Fifth and sixth columns: temporal summary. Last column: output label prediction.

**Figure 2 sensors-23-02750-f002:**
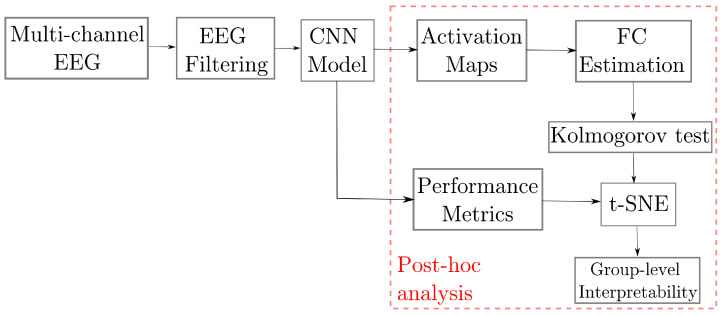
Guideline of the proposed framework for enhanced post-hoc interpretability of MI neural responses using connectivity measures extracted from EEGNet CAMs and clustering subjects, according to the EEG MI performance.

**Figure 3 sensors-23-02750-f003:**
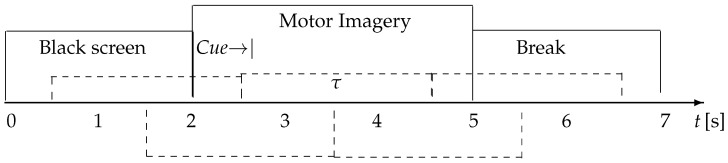
GigaScience database timeline of the evaluated motor imagery paradigm.

**Figure 4 sensors-23-02750-f004:**
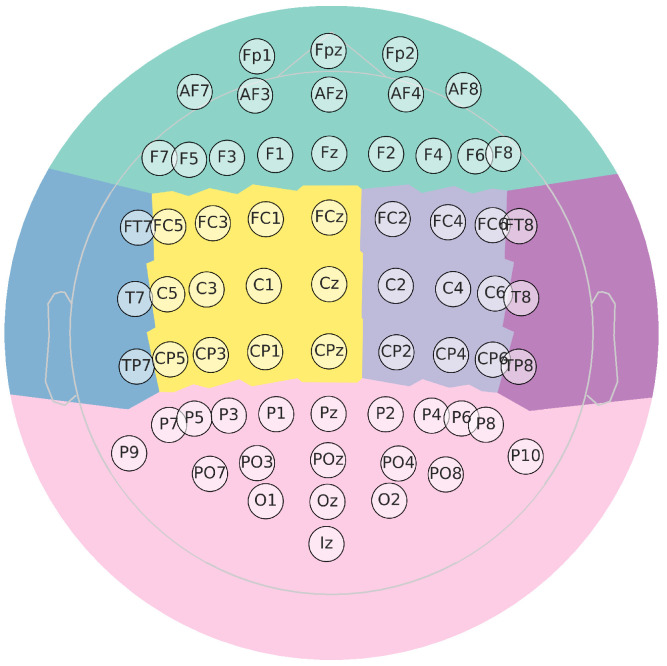
Topographic map for EEG representation. Besides, it highglights in color the main parts of the brain (Frontal, Central right, Posterior right, Posterior, Posterior left, Central left).

**Figure 5 sensors-23-02750-f005:**
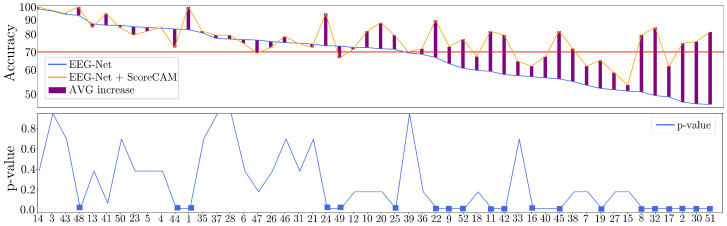
Subject-dependent MI discrimination results. (**Up**): Classification accuracy achieved for Bi-class MI tasks (left-hand and right-hand). Note: the red line at 70% level shows frequently used for fixing the poor MI coordination skill threshold under the subjects are considered as worse-performing. (**Bottom**): Obtained *p*-value per subject using the Nemenyi post-hoc test (square marker represents a *p*-value < 0.05).

**Figure 6 sensors-23-02750-f006:**
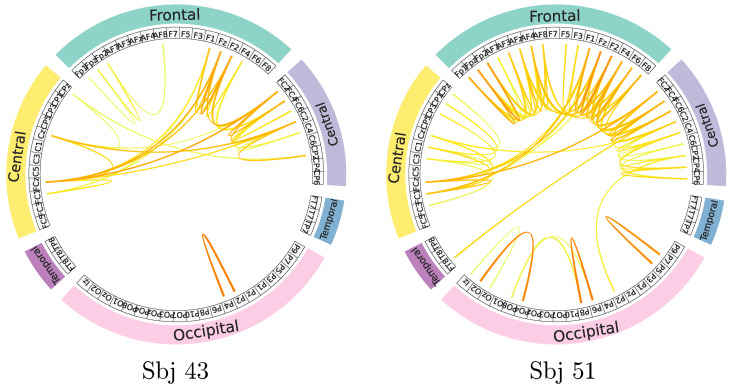
Connectivity maps are estimated for the best-performing (**left side**) and poorest-performing (**right side**) subjects. Stronger correlations between nodes are represented by darker edges linking two EEG channels. The right plot shows the brain regions (i.e., Temporal, Frontal, Occipital, Parietal, and Central) colored differently to improve spatial interpretation.

**Figure 7 sensors-23-02750-f007:**
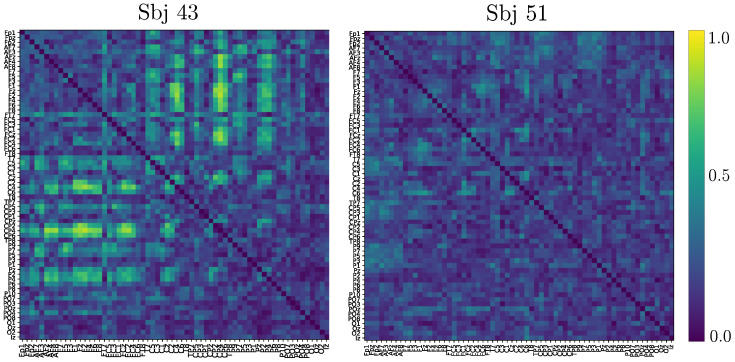
Connectivity matrix after two-sample Kolmogorov-Smirnov test obtained by the best (**left side**) and poorest-performing (**right side**) subjects. The pictured GFC matrices include both MI tasks and are computed at 90-percentile of normalized relevance weights.

**Figure 8 sensors-23-02750-f008:**
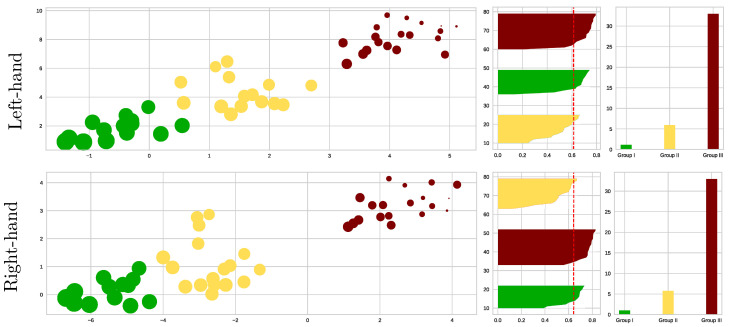
K−means clustering visualization of the EEG-Net classifier performance metrics and the Kolmogorov test values extracted from connectivity information. Left-side scatter plots of individuals (data points) estimated for the three considered groups of motor skills. Note that the point size represents the subject’s accuracy. The central plot shows the clustering metrics of the Silhouette coefficient, while the right-side plot displays the classifier performance, averaged across all subjects of each partition.

**Figure 9 sensors-23-02750-f009:**
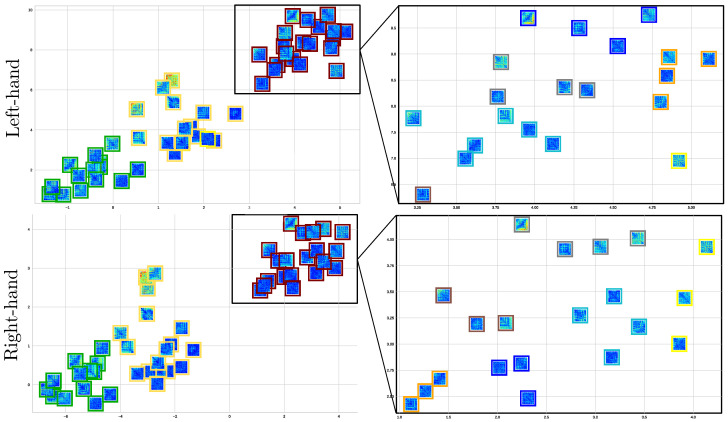
Clustering of motor skills obtained by the CNN-based approach with connectivity features for each MI task. (**Left side**): Subject partitions are colored for each subject group: GI (green squares), GII (yellow), and GIII (red). (**Right side**): Detailed analysis of the worst-performing GIII by splitting all the participants into subgroups, fixing k=6.

**Figure 10 sensors-23-02750-f010:**
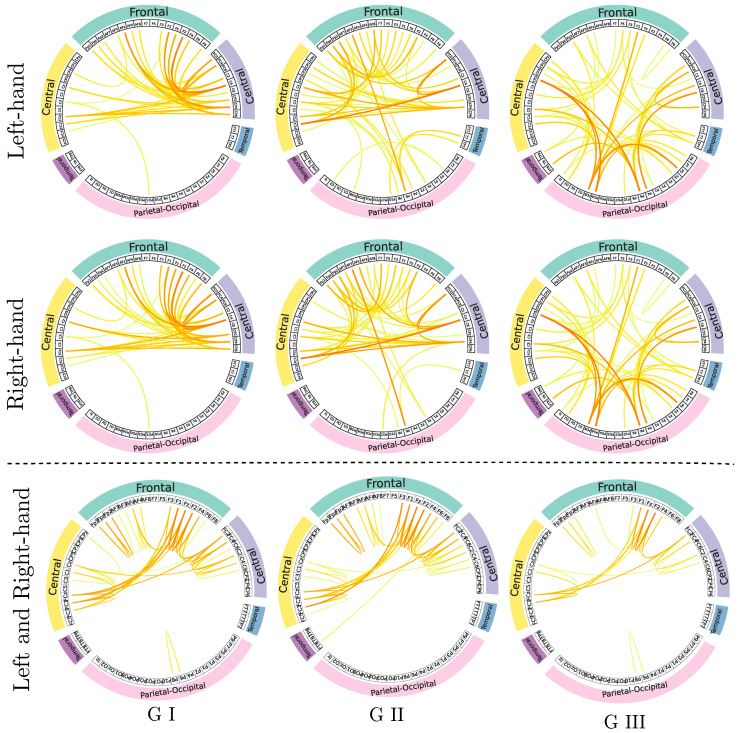
Graphical representation of the brain Gaussian Functional Connectivity estimated across each cluster of individual motor skills. Connectograms of each MI task (**top and middle rows**) are computed without the Komolgorov-Smirnov test and conducting the hypothesis test (**bottom row**). The GFC set is calculated at the 90 percentile of normalized relevance weights.

**Table 1 sensors-23-02750-t001:** EEGNet architecture. Input shape is (1,C,T), where *C* is number of channels, *T*—number of time points, $F_{1}$—number of temporal filters, *D*—depth multiplier (number of spatial filters), $F_{2}$—number of point-wise filters, and *N*—number of classes, respectively. For the Dropout layer, we use p=0.5 for within-subject classification. Notations ${}^{*}$ stand for $16*D*F_{1}+F_{2}*\left(D*F_{1}\right)$ and ${}^{\left(**\right)}$ for $N*(F_{2}*T\;/\;/32)$.

*Layer*	*Conv2D*	*Depthwise*	*Separable*	*Flatten*	*Dense*
*Name*		*Conv2D*	*Conv2D*		
# filters	$F_{1}$	$D*F_{1}$	$F_{2}$		${}^{\left(**\right)}$
Size	(1,64)	(C,1)	(1,16)		
# params	$64*F_{1}$	$C*D*F_{1}$	${(}^{*})$		
Output	($F_{1},C,T$)	($D*F_{1},1,T$)	($F_{2},1,T\;/\;/4$)	($F_{2}*\left(T\;/\;/32\right)$)	N
Options	Activation = Linear	Activation = Linear	Activation = Linear		
	Mode = same	Mode = same	Mode = same		
		Depth = *D*			
		max_norm = 1			
	BatchNorm = True	BatchNorm = True	BatchNorm = True		
		Activation = ELU	Activation = ELU		
		AvgPool2D = (1,4)	AvgPool2D = (1,8)		
		Dropout* − p=0.25	Dropout* − p=0.25		
		or p=0.5	or p=0.5		

**Table 2 sensors-23-02750-t002:** Comparison of bi-class accuracy achieved by state-of-the-art approaches in GigaScience collection. ${}^{*}$ The baseline approach (FBCSP+LDA) using conventional pattern recognition methods is also reported for comparison. The proposed CNN-based approach introducing spatiotemporal ScoreCAMs is competitive (marked in bold).

Approach	Accuracy	Interpretability
FBCSP+LDA ${}^{*}$	67.7 ± 13.6	–
LSTM+Optical [56]	68.2 ± 9.0	–
EGGnetv4+EA [57]	73.4	✓
DC1JNN [58]	76.50	✓
MINE+EEGnet [59]	76.6 ± 12.48	✓
Cropping+CNN [55]	86.5	✓
Pretrained ShuffleNet [53]	87.7	–
CSP+ST+CNN-PIM [54]	97.67 ± 2.0	–
EEGNet	69.7 ± 14.5	✓
EEGNet+ScoreCam (ours)	78.2 ± 10.9	✓

## Data Availability

Publicly available datasets were analyzed in this study. This data can be found here: http://gigadb.org/dataset/100295, accessed on 1 October 2022.

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
