# Peer review of "Posthoc Interpretability of Neural Responses by Grouping Subject Motor Imagery Skills Using CNN-Based Connectivity"

_sensors, 2023, doi:10.3390/s23052750_

Round 1
Reviewer 1 Report
In this paper, the authors presented a method for motor imagery classification. The major weakness of the paper is the novelty. A similar approach has been proposed in another paper published in the Sensor journal recently :
Caicedo-Acosta, J.; Castaño, G.A.; Acosta-Medina, C.; Alvarez-Meza, A.; Castellanos-Dominguez, G. Deep Neural Regression Prediction of Motor Imagery Skills Using EEG Functional Connectivity Indicators. Sensors 2021, 21, 1932. https://doi.org/10.3390/s21061932
The authors should compare their method and also compare their obtained results with the method and results presented in the above paper.
It is not clear whether the authors performed multiple comparisons for statistical analysis or not. This should be clarified in the revised version.
The quality of the figures should be improved.
The presentation of the equations should be double-checked by the authors. In the current version, some ambiguous symbols such as “!” is seen in some of the equations.
Reviewer 2 Report
1. Authors may revise the abstract to elaborate more on the problem statement, findings, and contributions.
2. Introduction is not clear. Authors may contribute more towards this.
3. Authors may elaborate more on the novelty/contribution of their work and how it contributes to the literature in the second last paragraph of the introduction clearly.
4. Authors need to be specific about their problem statement and the scope of their research.
5. Overall, the paper presentation requires improvement.
6. Thorough proofreading is recommended.
Reviewer 3 Report
Dear author,
Please find the attachment. I m recommending "Major Revision" of the study.

Round 2
Reviewer 2 Report
The authors addressed all the comments and concerns carefully; The manuscript stands for acceptance now.
Reviewer 3 Report
Accepted